# Direct observation of DNA target searching and cleavage by CRISPR-Cas12a

Yongmoon Jeon[1], You Hee Choi[2,3], Yunsu Jang [2,3], Jihyeon Yu[4,5], Jiyoung Goo[1,6], Gyejun Lee[1,6], You Kyeong Jeong[4], Seung Hwan Lee [7,10], In-San Kim[1,8], Jin-Soo Kim [7,9], Cherlhyun Jeong [1,6], Sanghwa Lee [2,3] & Sangsu Bae [4,5]

Cas12a (also called Cpf1) is a representative type V-A CRISPR effector RNA-guided DNA endonuclease, which provides an alternative to type II CRISPR–Cas9 for genome editing. Previous studies have revealed that Cas12a has unique features distinct from Cas9, but the detailed mechanisms of target searching and DNA cleavage by Cas12a are still unclear. Here, we directly observe this entire process by using single-molecule fluorescence assays to study Cas12a from *Acidaminococcus* sp. (AsCas12a). We determine that AsCas12a ribonucleoproteins search for their on-target site by a one-dimensional diffusion along elongated DNA molecules and induce cleavage in the two DNA strands in a well-defined order, beginning with the non-target strand. Furthermore, the protospacer-adjacent motif (PAM) for AsCas12a makes only a limited contribution of DNA unwinding during R-loop formation and shows a negligible role in the process of DNA cleavage, in contrast to the Cas9 PAM.

[1] Center for Theragnosis, Korea Institute of Science and Technology, Seoul 02792, South Korea. [2] Advanced Photonics Research Institute, Gwangju Institute of Science and Technology, Gwangju 61005, South Korea. [3] Cell Logistics Research Center, Gwangju Institute of Science and Technology, Gwangju 61005, South Korea. [4] Department of Chemistry, Hanyang University, Seoul 04763, South Korea. [5] Research Institute for Convergence of Basic Sciences, Hanyang University, Seoul 04763, South Korea. [6] KHU-KIST Department of Converging Science and Technology, Kyunghee University, Seoul 02447, South Korea. [7] Center for Genome Engineering, Institute for Basic Science, Seoul 08826, South Korea. [8] KU-KIST Graduate School of Converging Science and Technology, Korea University, Seoul 02841, South Korea. [9] Department of Chemistry, Seoul National University, Seoul 08826, South Korea. [10]Present address: National Primate Research Center, Korea Research Institute of Bioscience and Biotechnology, Chungcheongbuk-do 28116, South Korea. These authors contributed equally: Yongmoon Jeon, You Hee Choi, Yunsu Jang. Correspondence and requests for materials should be addressed to C.J. (email: che.jeong@kist.re.kr) or to S.L. (email: sanglee@gist.ac.kr) or to S.B. (email: sangsubae@hanyang.ac.kr)

CRISPR-Cas (clustered regularly interspaced short palindromic repeats and CRISPR-associated proteins) systems provide adaptive immune responses in bacteria and archaea[1–3]. These systems are presently divided into two major classes on the basis of their sub-components and function: Class 1 CRISPR-Cas members utilize multi-subunit effector complexes, whereas Class 2 members work with a single multidomain effector protein for DNA or RNA cleavage[4,5]. By virtue of their need for only a simple effector, Class 2 members such as type II Cas9 and type V-A Cas12a have been harnessed to achieve genome editing in various organisms.

The type II Cas9 derived from *Streptococcus pyogenes*, named SpCas9, is the most widely used Class 2 CRISPR-Cas system, but the type V-A LbCas12a from *Lachnospiraceae bacterium* and AsCas12a from *Acidaminococcus* sp. have been alternatively applied to gene editing[6–8]. Several features distinguish Cas12a from Cas9. For example, Cas9 requires two small RNAs, termed CRISPR RNA (crRNA) and trans-activating crRNA (tracrRNA), whereas Cas12a uses a single crRNA guide RNA. Additionally, Cas9 generates DNA double-strand breaks by producing blunt ends in the region upstream and proximal to guanine-rich protospacer-adjacent motif (PAM) sequences, whereas Cas12a induces staggered cuts in the region downstream and distal to thymine-rich PAM sequences. With respect to structure, Cas9 contains two different nuclease domains, HNH and RuvC, whereas Cas12a contains a RuvC-like endonuclease domain with a Nuc domain for DNA cleavage[9–14]. For Cas9, it is revealed that the non-target and target DNA strands are cleaved by the RuvC and HNH domains, respectively[15]. However, the lack of the HNH domain in Cas12a proposed that Cas12a may contain a single active site in the RuvC domain and recent study supported it[16]. Furthermore, in contrast to Cas9, Cas12a has the ability to process its own pre-crRNA into crRNA[17], facilitating multiplex genome editing[18]. Taken together, these distinguishing features may imply major differences between Cas9 and Cas12a during both target DNA searching and cleavage.

To date, several groups have revealed the molecular structure of Cas9 by using X-ray crystallography[10,11,15,19] and single-particle cryo-electron microscopy[20] and have also investigated DNA binding and cleavage mechanisms by using single-molecule fluorescence imaging[21–23] and high-speed atomic force microscopy (HS-AFM)[24]. However, compared with our knowledge about Cas9, the dynamic features of these processes in Cas12a are still unclear, although recently biochemical and structural studies have characterized Cas12a-mediated cleavage steps[12–14,16,25–27]. Here, we perform a single-molecule fluorescence imaging assay with a long DNA molecule (21 kb) to reveal the mechanism of target DNA searching and a single-molecule fluorescence resonance energy transfer (FRET) assay with a short DNA fragment (50 bp), which allows us to observe the DNA cleavage mechanism of AsCas12a in detail. With these two assays, we can directly observe the entire dynamic process comprising target searching, recognition, and cleavage by Cas12a ribonucleoproteins (RNPs).

## Results

**Facilitated diffusion of AsCas12a RNPs on a long DNA.** We first examined how an AsCas12a RNP searches for its cleavage site on a long DNA molecule. To observe this dynamic process directly in real time, we performed single-particle tracking experiments using total internal reflection fluorescence (TIRF) microscopy as depicted in Fig. 1a. We prepared 21 kbp-long DNA duplexes containing a unique cleavage site located ~4 kb from one end (Fig. 1b, Supplementary Table 1, and Methods)[28]. The long DNA duplex was then immobilized on a PEG (poly-ethylene

glycol)-coated quartz surface via biotin–streptavidin interactions in both ends of DNAs (Fig. 1a). To visualize the movement of AsCas12a–crRNA complexes on the DNA template, we labeled the 3′ end of the crRNA with Cy5 dye (the red signal in Fig. 1c) to detect AsCas12a RNPs and simultaneously used Sytox orange, a DNA intercalating dye (the green signal in Fig. 1c), to detect DNA templates.

When the AsCas12a–crRNA-Cy5 complexes were introduced into a detection chamber containing pre-attached DNA templates labeled with Sytox orange, we mainly observed AsCas12a–RNPs moving along the DNA (Supplementary Movie 1), implying that AsCas12a–RNP utilizes one-dimensional (1D) diffusion for DNA target searching. A representative kymograph of the entire process, comprising DNA target searching, recognition, and cleavage, is shown in Fig. 1c. In this sequence, one AsCas12a RNP first bound to DNA and briefly diffused along it. The RNP then stalled at the desired on-target site for a substantial period of time (~50 s), after which the DNA fragments shrank to each tethered site, indicating DNA cleavage. The distribution of DNA locations at which AsCas12a RNPs stably halted, ultimately inducing DNA cleavage, was well correlated with positions 0.2 or 0.8 relative to the length of the whole DNA molecule (Fig. 1d), strongly supporting the idea that AsCas12a RNPs recognized the on-target site after 1D diffusion and then cleaved the DNA. We also observed that the AsCas12a RNP promptly released the PAM-distal portion of the DNA strand (the left fragment of DNA in Fig. 1b) but continued holding the PAM-proximal portion of the DNA strand (the right fragment of DNA in Fig. 1b) after inducing DNA cleavage, in contrast to SpCas9 RNPs, which hold both DNA fragments for an extended period of time[21]. Statistically, among the 54 independent AsCas12a RNPs that ultimately induced DNA cleavage, we observed that about 93% showed obvious 1D diffusion on the DNA before stable binding at the target site, as determined within the limits of our spatial and temporal resolution. This result strongly suggests that AsCas12a RNPs do in fact utilize 1D diffusion for facilitated target searching.

We further examined the relationship between the PAM density and the initial binding position of diffusing AsCas12a RNPs. We found that the initial binding position of AsCas12a RNPs before they began 1D diffusion was positively correlated with the PAM density ($r = 0.76$, $p < 0.001$, Pearson's linear correlation coefficient, Fig. 1e), indicating that AsCas12a RNPs have a preference for PAM sites before beginning diffusion.

**Molecular mechanism of AsCas12a RNP 1D diffusion.** To determine the detailed 1D diffusion mechanism of AsCas12a RNPs during the DNA target searching process, we further examined the movements of each AsCas12a RNP by implementing FIONA (fluorescence imaging with one nanometer accuracy)-based particle tracking with 2D Gaussian fitting of intensity profiles[29,30]. From the trajectory of each AsCas12a RNP, we obtained diffusion coefficients by using mean squared displacement (MSD) analysis. Under physiological conditions (100 mM KCl), the diffusive motion of the AsCas12a RNP was relatively fast ($D = 1.73 \pm 0.14$ $\mu m^2$/s, mean ± s.e.m.) compared with other DNA-binding proteins showing 1D diffusion (ex. ~0.087 $\mu m^2$/s for MutS ATP clamp[30], ~0.03 $\mu m^2$/s for P53[31], and ~0.02 $\mu m^2$/s for UL42[32]). However, the diffusion coefficient of the AsCas12a RNP decreased dramatically as the ionic strength decreased to 10 mM KCl (Fig. 1f). We found that AsCas12a RNPs move three times more slowly on the DNA duplex when the ionic strength was decreased tenfold, suggesting that ionic screening may affect the interaction between Cas12a and DNA. It also indicates that the AsCas12a RNP does not bind to DNA tightly

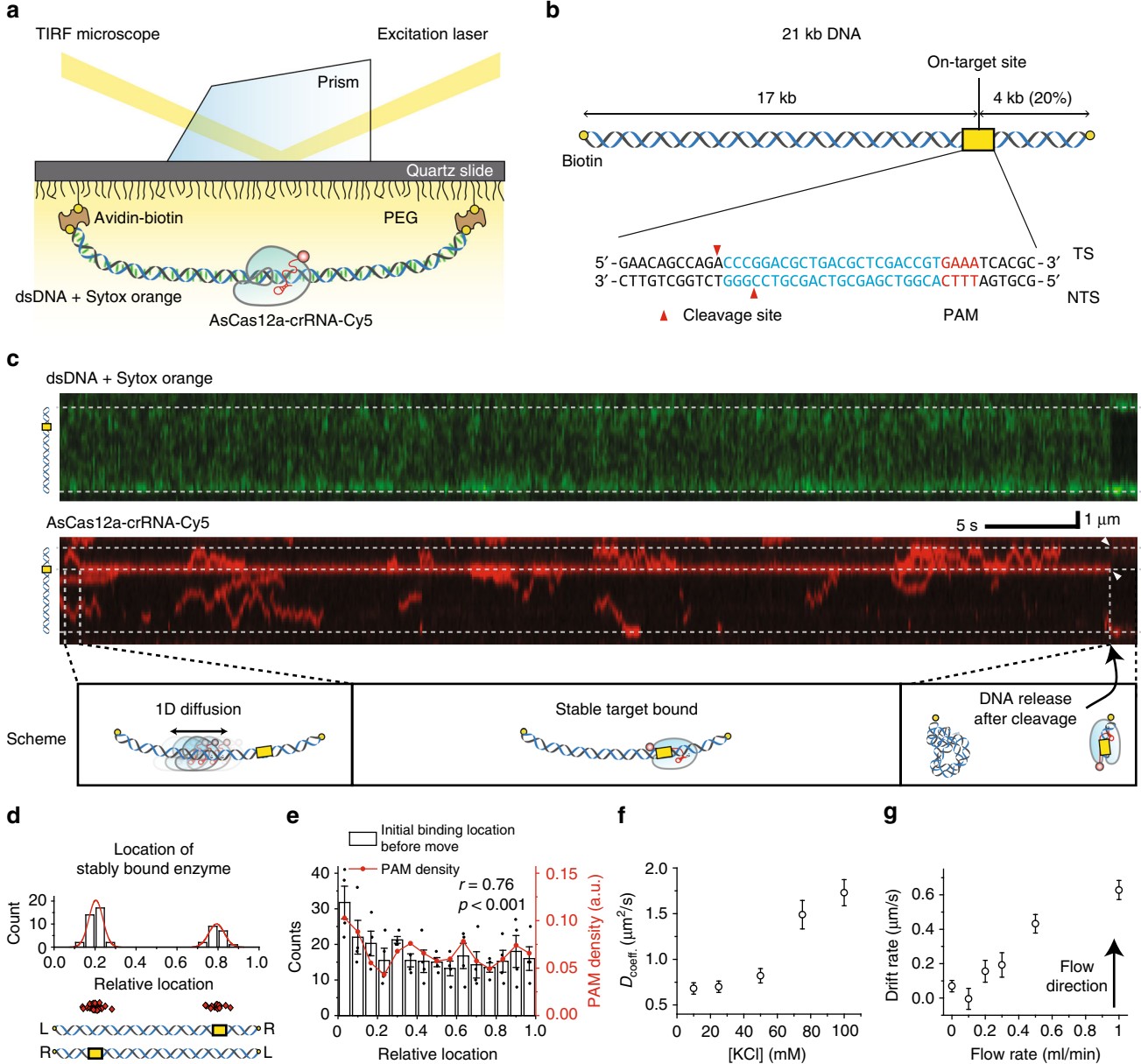

**Fig. 1** Real-time observation of AsCas12a RNP undergoing 1D diffusion for target searching. **a** Schematic representation of the TIRF microscope setup and the experimental design. **b** Diagram of the 21 kb DNA molecule containing biotins at both ends for immobilization. The AsCas12a–crRNA complex target site is located at a position that is 20% along the length of the DNA. DNA sequences around the target site are shown below. **c** Kymograph of dsDNA (visualized with Sytox orange) and AsCas12a localization (visualized with crRNA-Cy5) over time (x axis). **d** Position distribution of AsCas12a stably bound to the target. Two possible orientations of DNA containing the target site are shown below. (n P54; number of analyzed molecules). **e** The distribution of AsCas12a 1D diffusion start sites along the DNA (white bars) and the probability density of PAM sequences in a 21 kb DNA molecule (red line). The DNA orientation was determined by the location of a stably bound AsCas12a molecule. The degree of correlation between the two graphs was evaluated by Pearson's correlation analysis (r = 0.75, p < 0.001). Initial binding locations of AsCas12a were collected from at least four independent experiments (total number of data is 1054). **f** Ionic strength-dependent diffusion coefficients of AsCas12a RNP on the 21 kb DNA in the absence of external force. At least 191 molecules are analyzed for this graph from two independent experiments (236 for 10 mM; 233 for 25 mM; 291 for 50 mM; 191 for 75 mM; and 236 for 100 mM). **g** Flow rate versus drift rate of AsCas12a RNP on a single tethered λ-phage DNA. As the flow rate is increased, the AsCas12a RNP drift rate increases. At least 240 molecules are analyzed for this graph from two independent experiments (291 for 0 ml/min; 309 for 100 ml/min; 468 for 200 ml/min; 433 for 300 ml/min; 240 for 500 ml/min; and 248 for 1000 ml/min). The error bars in these figures mean the standard error of the mean

and diffuses by intermittent contact with DNA[33]. In a 1D sliding mechanism model, protein complexes continuously contact with DNA and follow its helical path so that diffusion coefficients do not vary with the ionic strength. This intermittent DNA contact is consistent with the observation that several AsCas12a RNPs frequently passed by a target site while DNA target searching (Supplementary Movie 1). In addition, as the flow rate increased, the AsCas12a RNP drift rate, which is the degree to which the RNPs are pushed out away from their locations on the DNA by hydrodynamic flow (Methods), was accordingly increased (Fig. 1g), supporting the fact that AsCas12a RNP does not maintain the continuous and tight contact with DNA.

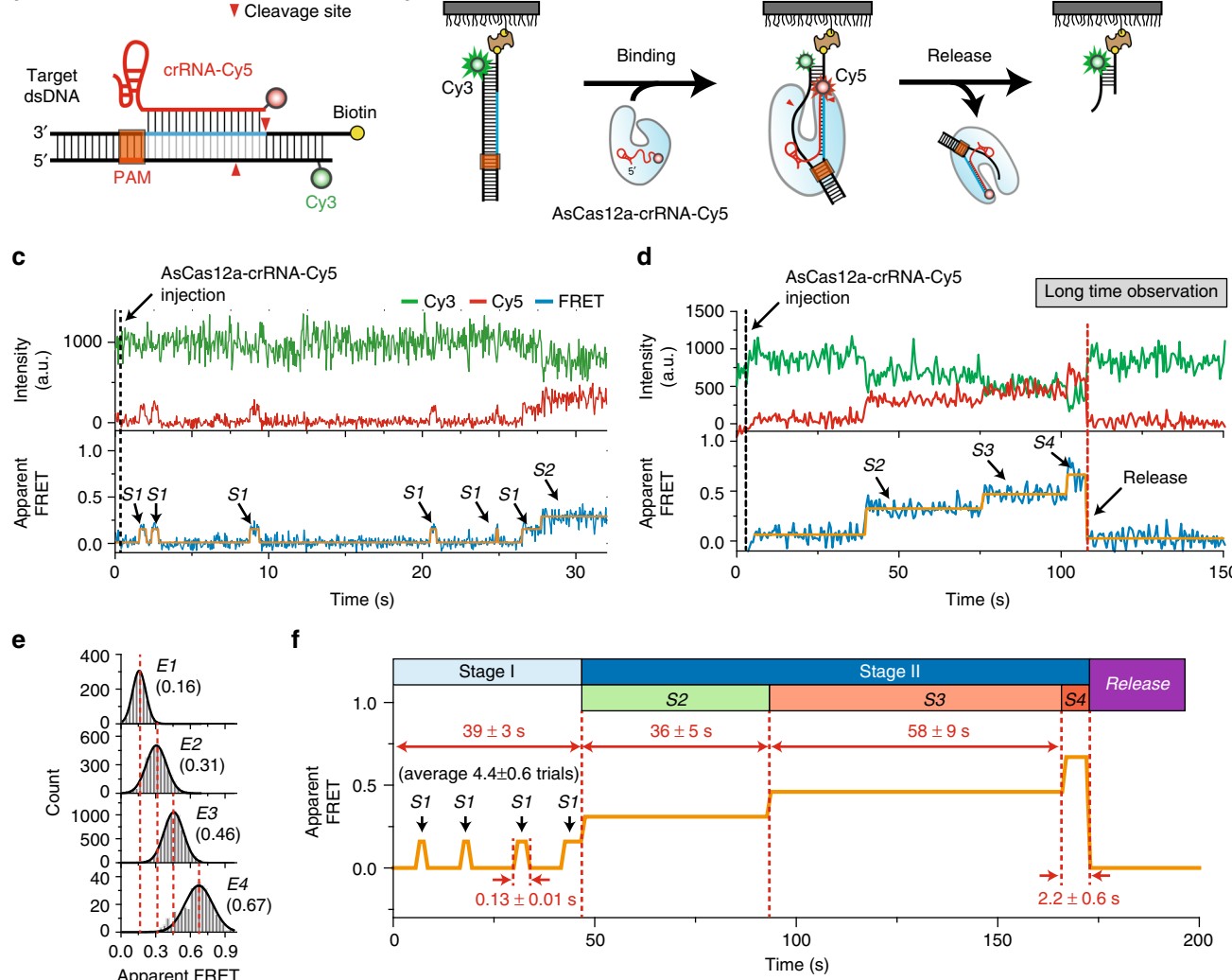

**Fig. 2** Direct observation of DNA cleavage by AsCas12a RNP. **a** The standard design of target DNAs and crRNAs used in the experiments. **b** Schematic diagram of the single-molecule FRET experiment used to monitor the AsCas12a cleavage reaction. **c**, **d** Representative time traces of Cy3 fluorescence (green, top), Cy5 fluorescence (red, top), and corresponding FRET efficiency (blue, bottom) showing the short-lived weak binding of AsCas12a RNP to target DNA (**c**) and the stable binding with subsequent FRET changes (**d**). To clearly visualize transitions, orange lines were added as an eye guide. The same color conventions are used throughout the paper. These experiments were performed under the same reaction conditions except for the time resolution. In **c**, a time resolution of 50-ms was used to clearly detect the short-lived binding events. In **d**, a time resolution of 500-ms was used instead to reduce the photobleaching of the fluorophores for long-time observation. **e** FRET histograms of individual reaction steps. FRET values for each step were obtained by fitting the FRET histograms to a single Gaussian function. To obtain the FRET histograms, all data points for each state were collected from at least more than 45 time trajectories (65 for E1; 63 for E2 and E3; and 45 for E4). **f** Representative scheme of a time trace showing both short- and long-lived binding events and the subsequent FRET changes accompanying the final release of the cleavage product

**Direct observation of DNA cleavage by AsCas12a RNPs.** We next investigated the DNA cleavage reaction induced by AsCas12a RNPs after target searching. To observe the interactions between the target DNA and AsCas12a RNPs in real time, we performed single-molecule FRET experiments. As shown in Fig. 2a, we prepared partial DNA duplexes labeled with a FRET donor (Cy3) and used crRNAs with a FRET acceptor (Cy5). DNA substrates were immobilized on a PEG-coated quartz surface as in the experiment described above (Supplementary Table 1 and Methods). Preassembled AsCas12a–crRNA-Cy5 complexes were then delivered into the detection chamber while fluorescence signals from single DNA and RNA molecules were monitored using a TIRF microscope (Fig. 2b).

Representative traces from two independent experiments with different time resolutions (50 ms and 0.5 s) are shown in Fig. 2c, d, respectively. Upon the addition of AsCas12a RNPs, we detected

target DNA-binding events by the appearance of the Cy5 signal and FRET changes. In this experiment, we measured two distinct stages with different binding modes: stage I, in which AsCas12a RNPs repeatedly bind to target DNA with a short lifetime of 0.13 s (indicated by S1 in Fig. 2c), and the subsequent stage II, in which the AsCas12a RNP and target DNA ternary complex is stably maintained and has a much longer lifetime of 94 s (indicated by S2–S4 in Fig. 2d). We also found that the stable S2 state was formed after average of 4.4 transient S1-binding trials (Supplementary Figure 1); subsequent steps, including DNA cleavage and release, occurred during the long-lived states in stage II. In our previous study of SpCas9 RNPs[22], the short-lived binding events were not apparent, possibly indicating that the stability of an initial binding of AsCas12a RNPs to a target DNA might be weaker than that of SpCas9 RNPs.

In stage II, the transient-binding state (S1) proceeded to subsequent steps through three distinct FRET states (S2, S3, and S4 in Fig. 2d); the Cy5 signal from the crRNA eventually disappeared during step S4, indicating crRNA release. In this measurement, the disappearance of Cy5 signal can be also interpreted as Cy5 photobleaching, but we excluded this possibility based on the fact that the dye photobleaching time (~1500 s) under the same experimental condition was about 15 times longer than the measured release time (~100 s) (Supplementary Figure 2). To confirm that crRNA release results from DNA cleavage, we performed the same experiment with doubly labeled target DNA, which allows us to observe release from the cleaved DNA fragment directly (Supplementary Figure 3). In these studies, we found that the cleaved DNA and the AsCas12a RNP were simultaneously released after DNA cleavage, which is consistent with the results from the long DNA molecule described above (Fig. 1c).

FRET histograms (Fig. 2e) and kinetic data (Supplementary Figure 4) for individual reaction states established the existence of four structurally and kinetically distinct ternary complex configurations. Based on these measurements, we conclude that the AsCas12a RNP-mediated DNA cleavage reaction occurs through four distinct and well-ordered reaction steps as summarized in Fig. 2f. Furthermore, it is notable that most S2 events proceed to subsequent states including target cleavage and release of cleaved DNA, suggesting that the formation of the stable-binding complex (S2) is a key step for target recognition as a stable R-loop conformation.

**Defining individual reaction steps with mutants**. We next assigned the subsequent FRET intermediates to structural and catalytic intermediates in stage II by studying the behavior of altered components. When we employed a catalytically deactivated AsCas12a, named dAsCas12a, which has an E993A mutation in the RuvC active site, the stable binding events occurred but further events, including DNA cleavage, did not. Instead, the complex remained at S2 as shown in Fig. 3a. In addition, when a nick was introduced in the scissile bond of the target strand (termed TS-nick) and wild-type (WT) AsCas12a was used, cleaved DNA was promptly released during the S2 state, but the S3 and S4 states were not detected (Fig. 3b). Based on these two observations, we conclude that S2 represents a pre-cleavage complex and that the first cleavage event occurs in the non-target strand during S2.

On the other hand, when a nick was introduced in the scissile bond of the non-target strand (termed NTS-nick) and WT AsCas12a was used, the stage II complex started from the S3 state, not the S2 state, and then proceeded to the remaining steps (Fig. 3c). This result strongly suggests that S3 corresponds to a product of the first cleavage event, which occurred on the non-target DNA strand. In addition, when both dAsCas12a and NTS-nick DNA were used (Fig. 3d), the complex in the S3 state did not proceed to the remaining steps, implying that S3 also represents a pre-cleavage state that facilitates cleavage of the target strand and that the complex at the remaining step (S4) may be a product of the second cleavage before the release of cleaved DNA. Furthermore, it is notable that, when WT AsCas12a was used, the S4 state was observed for NTS-nick DNA before the release of cleaved DNA, but not for TS-nick DNA. Considering that the cleavage of target strand is not necessary in TS-nick DNA, this difference may reflect that a large conformational change of DNA is required for the cleavage of target strand. This assumption can be advocated by a recent study about Cas12b crystal structure that the target DNA strand was kinked to access the catalytic site in the RuvC domain for target strand cleavage[34].

The finding that deactivation at the catalytic residue (E993A) in the RuvC domain impairs target strand cleavage even in the presence of a nick in the non-target strand conflicts with the earlier argument that non-target strand cleavage by the RuvC domain is a prerequisite for cleavage of the target strand by Cas12a[14]. Rather, it seems likely that the RuvC domain may directly catalyze target strand cleavage[16]. Consequently, we conclude that the two DNA strands are sequentially cleaved by the single catalytic site in the RuvC domain in a defined order, beginning with the non-target strand and progressing to the target strand.

**Minimal motif for stable binding of AsCas12a to DNA**. Previous studies reported that the specific targeting of both Cas9 and Cas12a to DNA is dominantly influenced by PAM-proximal sequences[12,15,16,23]. In the case of SpCas9 RNPs, a perfect match between the crRNA and the 9–10 nucleotide (nt) region proximal to the PAM secured sufficient stability for target recognition[23]. Similar to Cas9, it has been reported that Cas12a utilizes seed-dependent DNA targeting[16].

To investigate the correlation between the length of matched PAM-proximal sequences and formation of the stable R-loop conformation, we employed a series of DNA substrates with varied lengths of crRNA-matched base pairs (bps) extending from the PAM-proximal side as shown in Fig. 4a. We used dAsCas12a in these assays to eliminate any possible interference from subsequent cleavage and release events and increase the reliability of our data analysis. Figure 4b shows representative fluorescence intensity time traces displaying AsCas12a RNP-binding events with the series of DNA substrates. Kinetic analysis revealed that stable binding events (stage II in Fig. 3a) required 17 nt of PAM-proximal sequence to match with the crRNA sequence. This result means that AsCas12a RNPs need a relatively longer stretch of matched nucleotides for stable R-loop formation compared with SpCas9 RNPs, which only require 9–10 nt matches[23]. We further performed an in vitro DNA cleavage assay using an agarose gel electrophoresis with WT AsCas12a and DNA plasmids containing the same target sequences (Supplementary Table 2 and Methods) to the smFRET experiments to investigate the correlation between the binding lifetime and the DNA cleavage efficiency (Supplementary Figure 5). Consequently, we found that the cleavage efficiency was positively correlated with the binding lifetime and that it likewise significantly increased when the length of matched DNA reached 17 nts (Fig. 4c). Based on these observations, we conclude that formation of at least 17 bps of an RNA–DNA heteroduplex is a prerequisite for stable AsCas12a RNP binding and subsequent cleavage of DNA, which is consistent with previous findings that crRNA–DNA pairings in the PAM-distal region are of negligible importance for DNA cleavage by Cas12a[12,35,36] and a recent single-molecule study[37]. In addition, it is notable that the DNA construct lacking any match with the crRNA (zero matched nucleotides in Fig. 4c) showed no observable-specific binding events in our time resolution ($t < 30$ ms), suggesting that Cas12a RNPs have weaker bind affinity compared to Cas9 RNPs, which bind to such a fully mismatched DNA construct for ~1s[23].

**A PAM is required only for R-loop initiation**. To investigate the role of PAM sequences during Cas12a target searching and DNA cleavage, we introduced non-matched sequences (a bubble formation) at the target sequence of DNA templates with or without a PAM as depicted in Fig. 5a. We revisited the single-molecule FRET assay with AsCas12a RNPs and measured the dwell time of each stage as depicted in Fig. 5b. We first compared the association time for making a stable R-loop complex (i.e., the dwell

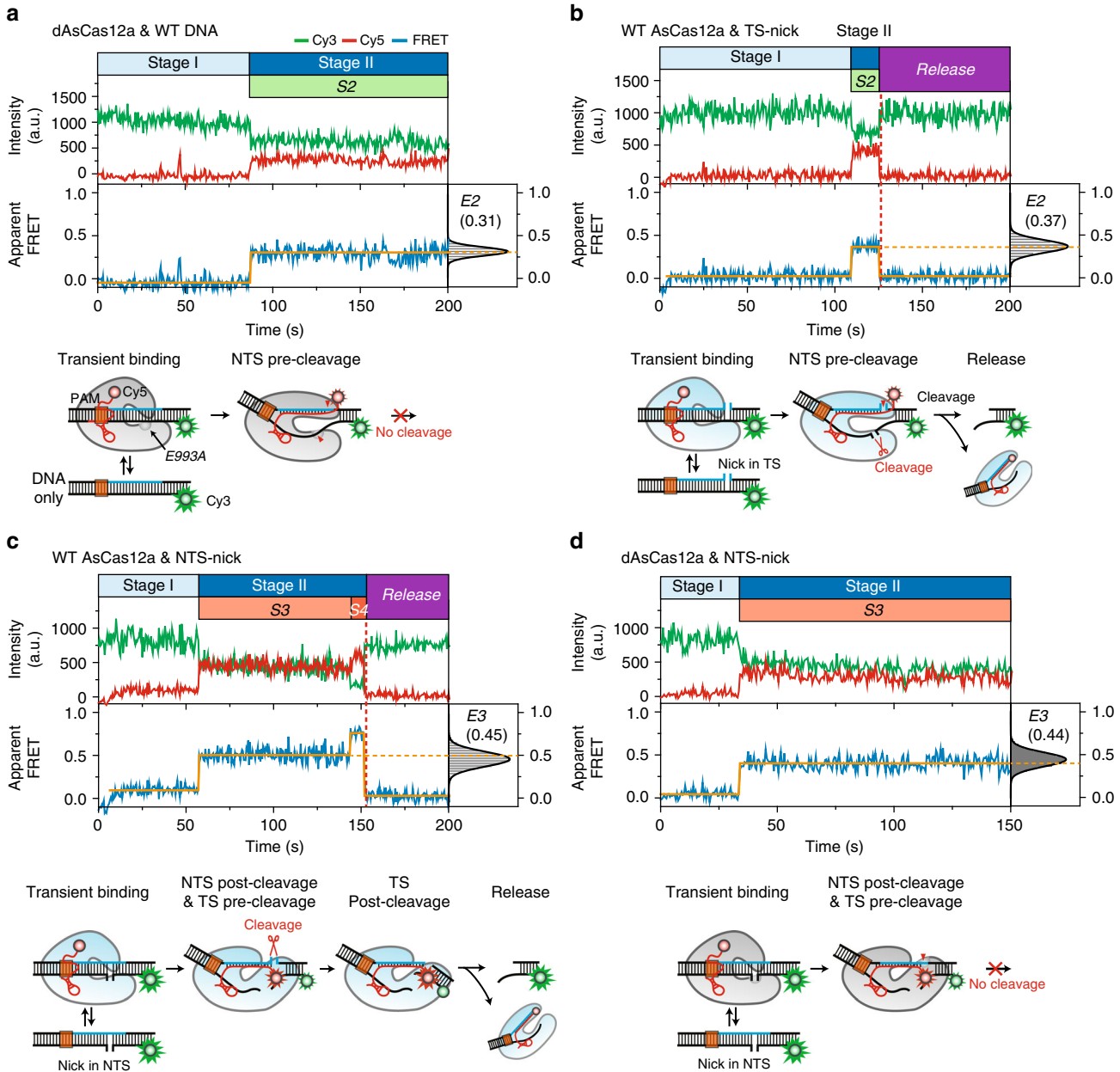

**Fig. 3** Direct observation of sequential double-stranded DNA cleavage by AsCas12a RNP. Representative time traces of fluorescence intensity (top) and FRET efficiency (bottom) for WT DNA with catalytically deactivated dAsCas12a (E993A) (**a**), TS (target strand)-nicked DNA with WT enzyme (**b**), NTS (non-target strand)-nicked DNA with WT enzyme (**c**), NTS-nicked DNA with dAsCas12a (E993A) (**d**) in the presence of 20 nM enzyme and 20 nM crRNA. FRET histograms of individual reaction steps were added to allow direct comparison of each state. To obtain the FRET histograms, all data points for each state were collected from at least more than 46 time trajectogires (125 for **a**; 50 for **b**; 46 for **c**; and 113 for **d**). Experimental schemes to show the state assignment of FRET intermediates were added below each time trace

time of stage I) for the three DNA constructs. Surprisingly, AsCas12a RNPs formed an R-loop complex with the DNA with a bubble structure about nine times faster than with WT DNA, regardless of the presence of the PAM sequence (Cyan bars in Fig. 5c). In contrast, SpCas9 RNPs could not form a stable R-loop complex when the PAM sequences were absent, even though the protospacer was already unwound. These observations indicate that AsCas12a RNPs have a relatively weaker binding affinity with PAM sequences than do SpCas9 RNPs. The result does not mean that the PAM is unnecessary for AsCas12a RNPs during the target searching process because AsCas12a cannot bind to target DNAs that lack both a PAM sequence and a bubble formation (Supplementary Figure 6).

We next compared the cleavage time (i.e., the dwell time of stage II) for the three DNA constructs and found that the reaction times were relatively similar (Fig. 5d). This result indicates that once the R-loop is formed, the presence of a PAM does not affect cleavage dynamics. In other words, the role of the PAM for AsCas12a RNPs is limited to unwinding the DNA base pairing for R-loop initiation; there is no role in the cleavage reaction steps after R-loop formation is complete.

## Discussion
In this study, we directly observed the entire interaction of AsCas12a with DNA, from target searching to DNA cleavage, at

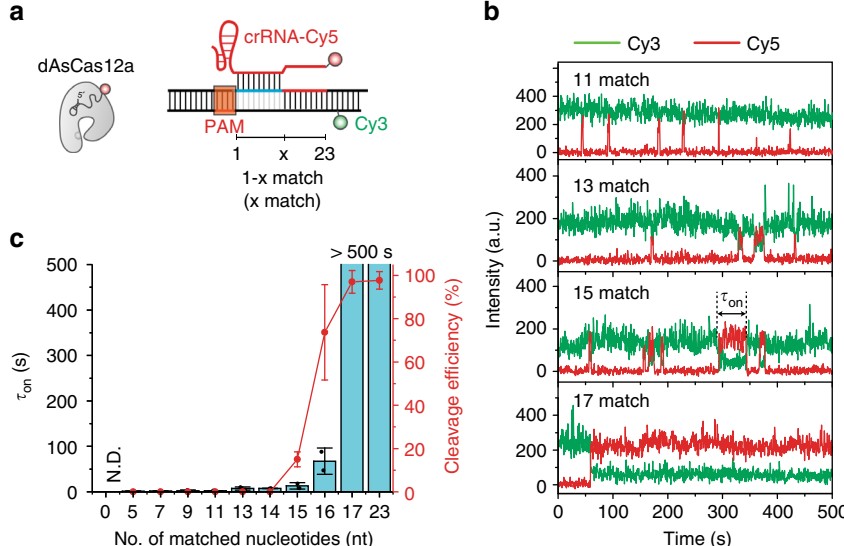

**Fig. 4** Stability of AsCas12a on partially cognate DNA templates. **a** Diagram of designed Cy3-labeled DNA templates and Cy5-labeled crRNA. Part (1−x nt) of the protospacer is complementary to the crRNA. Catalytically deactivated dAsCas12a was used to avoid additional complexity from DNA cleavage activity. **b** Representative time traces of Cy3 and Cy5 fluorescence intensity that show binding and unbinding activity of AsCas12a on each DNA template (with 11, 13, 15, 17 matched nucleotides). AsCas12a-binding results in FRET. **c** The dAsCas12a dwell time and cleavage efficiency versus the number of matched nucleotides. The dwell time is represented by blue bars and the cleavage efficiency by red dots. The dwell time ($\tau_{on}$) was extracted from time traces as in **b** and fitted with a single exponential decay function. At least 111 time traces are analyzed from two independent experiments (254 for 0 match; 495 for 5 match; 333 for 7 match; 343 for 9 match; 318 for 11 match; 401 for 13 match; 206 for 14 match; 325 for 15 match; 317 for 16 match; 201 for 17 match; and 111 for 23 match). The cleavage efficiency was assessed by in vitro cleavage assay using agarose gel electrophoresis (Supplementary Figure 5). The error bars mean the standard deviation ($n = 3$). All FRET experiments in this figure were performed under the same reaction conditions (10 nM AsCas12a, 20 nM crRNA, 10 mM Tris-HCl (pH 8.0), 100 mM KCl, 10 mM MgCl$_2$, and 1 mM DTT)

the single-molecule level. We first observed direct movements of single AsCas12a RNPs and confirmed a 1D hopping mechanism for target searching on the long DNA molecules. We next observed the nature and dynamics of processes ranging from target recognition to DNA cleavage, sequentially and in detail, on short DNA fragments. Based on these observations, we propose a model for AsCas12a as summarized in Fig. 6. In this model, AsCas12a RNPs bind to DNA and start to move along it by a hopping mechanism, sometimes passing through a target site. If the AsCas12a RNPs encounter a potential target site containing a PAM sequence, they begin to initiate R-loop formation by forming base-pair hybrids between the crRNA and the target DNA strand. During this step, matching sequences in the PAM-proximal region are crucial; AsCas12a–crRNA–DNA ternary complexes rarely maintain an R-loop formation if the number of matched nucleotides is insufficient (<17 bp of matching target sequence). If sufficient matched nucleotides exist, AsCas12a RNPs establish a stable R-loop formation and proceed to an active state for cleaving DNA. After a stable R-loop conformation is formed (pre-cleavage state, S2 in Fig. 2f), AsCas12a, via the activity of its RuvC domain, first cleaves the non-target DNA strand (first cleaved state, S3 in Fig. 2f) and then cleaves the opposite, target DNA strand (second cleaved state, S4 in Fig. 2f), regardless of the presence of a PAM sequence. In contrast to Cas9, which does not release either DNA fragment after cleavage[21,22], AsCas12a rapidly releases the PAM-distal DNA fragment but continues to hold the PAM-proximal portion of DNA.

Facilitated diffusion, which is a combination of 3D bulk diffusion and 1D diffusion along the DNA, is known to be a common strategy for DNA-binding proteins to find specific target sites efficiently[30,32,38–42]. Nevertheless, to the best of our knowledge, no proof of a 1D diffusion mechanism during target searching has been reported for the CRISPR-Cas12a. In this

study, we observed that the AsCas12a RNP shows 1D diffusion by intermittent contact on DNA and that the initial binding position of diffusing AsCas12a RNP are positively correlated with the PAM density. These results imply the importance of PAM recognition by AsCas12a for 1D diffusion. Recent structural studies[13,26,27] reported that both LbCas12a and AsCas12a RNP have large conformational changes before and after binding on-target DNA. In particular, Gao et al.[27] proposed that the PAM interaction of AsCas12a RNP triggers the entire structural rearrangement of itself. It may support our description that AsCas12a can form 1D diffusion compatible conformation through PAM recognition.

Type II Cas9 contains RuvC and HNH domains, which are responsible for cleavage of the non-target and target DNA strands, respectively[15]. In contrast, the type V-A CRISPR effector proteins, including Cas12a and Cas12b (also called C2c1), contain a RuvC domain and a Nuc domain, rather than the HNH domain of Cas9[12,43]. In Cas12 enzymes, the non-target strand is reported to be cleaved by a RuvC domain similar to the process in Cas9, but there is still controversy regarding target strand cleavage. Yamano et al.[26] suggested that the target strand may be cleaved by the Nuc domain, although cleavage of the non-target strand by the RuvC domain is a prerequisite for target strand cleavage, based on mutational studies of putative catalytic residues located in the RuvC and Nuc domains. However, two recent structural studies[16,34] suggested that both Cas12a and Cas12b contain a single catalytic site in the RuvC domain, which is responsible for the cleavage of both target and non-target DNA strands. In particular, the Cas12b crystal structure directly showed that the catalytic residues located in the RuvC domain, which are highly conserved in Cas12a, interact with the cleavage site in the target strand[34]. Consistent with this conclusion, we found that mutation of the catalytic residues in the RuvC domain impairs cleavage of the target strand even in the presence of a nick in the scissile bond

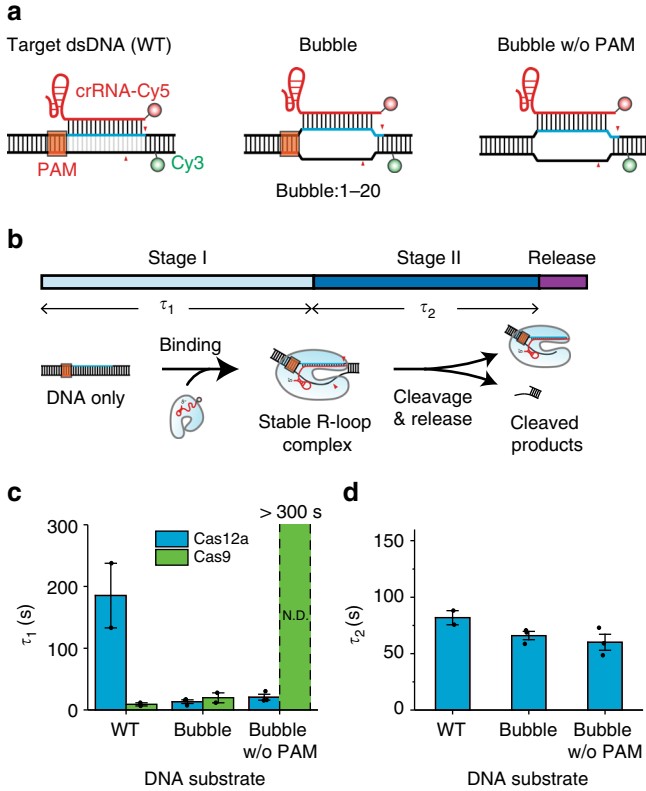

**Fig. 5** The PAM serves a limited role for AsCas12a function. **a** Diagram of DNA templates. Target dsDNA (WT) (left) contains a PAM and a proper protospacer. Bubble DNA (middle) has a PAM and a bubble structure (mismatches between the target and non-target strands) in the protospacer. Bubble w/o PAM (right) is the same as Bubble DNA except that it lacks a PAM sequence. Cy3 and biotin modifications are identical for all three DNA templates. **b** Schematic diagram of the single-molecule FRET experiment. **c, d** Target binding time $\tau_1$ (**c**) and cleavage time $\tau_2$ (**d**) of the AsCas12a–crRNA complex for the three different DNA templates. At least 171 time traces are analyzed to obtain $\tau_1$ and $\tau_2$ from two independent experiments (for Cas12a: 214 for WT, 171 for Bubble, 271 for Bubble w/o PAM, for Cas9: 404 for WT, 410 for Bubble, 396 for Bubble w/o PAM) (mean ± s.e.m.). All values in this figure are measured with a same concentration; 10 nM of protein (AsCas12a or SpCas9) and 20 nM RNA (crRNA or crRNA-tracrRNA)

of the non-target strand (Fig. 3d). This observation implies that simple cleavage of the non-target strand is not a prerequisite for cleavage of the target strand, but that the RuvC domain may directly catalyze target strand cleavage.

On the other hand, it has not yet been clarified whether the two DNA strands are sequentially cleaved in a defined order. In this study, we found that Cas12a cleaves the non-target and target DNA strands in a sequential manner, beginning with the non-target strand. Furthermore, we found that a significant FRET signal increase occurred after the first cleavage event in the non-target strand (from 0.31 during S2 to 0.46 during S3 in Fig. 2e). The structural comparison between Cas12a and Cas12b showed that the distance between the putative site of the target strand cleavage and the catalytic site in the Cas12a RuvC domain is considerably longer than that in Cas12b, which suggests that additional movements of the Cas12a catalytic RuvC domain toward the target strand are required to facilitate target strand cleavage[16,34]. Therefore, it is plausible that the conformational change in the R-loop complex leads to target strand cleavage, causing the increase in the FRET signal.

In the CRISPR-Cas system, the PAM sequence in the target DNA is critical for distinguishing self from non-self DNA;[44,45] in general, CRISPR effector proteins bind target sites with the help of the PAM. However, it is suspected that the role of the PAM differs for various types of CRISPR endonucleases. In the case of type I cascade, there is an additional process during DNA cleavage of checking the PAM sequence after R-loop-zipping occurs, indicating that the PAM site has a role in determining the target site during DNA cleavage[46]. In this study, we observed that Cas12a proceeds to the cleavage process regardless of the presence of a PAM if the target sequence is already unwound (Fig. 5c) and that the absence of a PAM does not affect DNA cleavage (Fig. 5d). In contrast, Cas9 cannot strongly bind to the target DNA without PAM recognition in the presence of bubble DNA. Based on these observations, we tentatively argue that, unlike type I cascade and type II Cas9, the PAM of Cas12a plays a negligible role after DNA unwinding for R-loop initiation. Our study also showed that a 17-bp seed is required for stable Cas12a binding (Fig. 4), which is longer than the 7–8-bp seed requirement of type I cascade[47] and the 9–10-bp seed requirement of type II Cas9[23]. The longer seed sequences for stable binding of AsCas12a RNP may provide insight into its higher target specificity;[36,48] similarly, in a previous study, Cas9 specificity was improved compared to that of wild-type Cas9 by reducing its binding affinity for target DNA[49,50]. Thus, our results may give insight to elucidate the basic properties of various types of CRISPR enzymes and to promote efficiency and specificity in genome editing.

## Methods

**Protein expression and purification**. *E. coli* expressed recombinant AsCas12a and SpCas9 plasmids; pET28b-AsCas12a (WT), pET28b-deadAsCas12a (E993A), and pET28b-SpCas9 were provided from Prof. Jin-Soo Kim. Competent BL21-Pro cells (CP111, Ezynomics) were transformed with recombinant protein plasmids and single colonies were obtained on an agar plate containing kanamycin. A single colony was inoculated to LB medium containing kanamycin and grown at 37 °C with shaking at 200 rpm to mid-log phase (OD$_{600}$ nm ~0.4–0.5). A protein expression was induced with 0.8 mM IPTG and incubate the cells overnight at 18 °C with shaking at 200 rpm. Cells were collected by centrifuging for 20 min at 3100 × $g$ and 4 °C and resuspended in lysis buffer (50 mM NaH$_2$PO$_4$, 300 mM NaCl, 10 mM imidazole). A lysozyme and 1 mM PMSF were added to lysate and incubated for 1 h on ice. Cells were sonicated (amplitude 25%, 1 s ON/1 s OFF, pulse 30 times) repeatedly for five times and the lysate was cleared by centrifugation 4 °C 18,000 × $g$, 30 min. The supernatant was filtered through 0.45-μm syringe filters and was incubated with Ni-NTA agarose for 1 h with rotation at 4 °C. The resin was washed for two times with wash buffer (50 mM NaH$_2$PO$_4$, 300 mM NaCl, 20 mM imidazole). The bounded proteins were eluted with elution buffer (50 mM NaH$_2$PO$_4$, 300 mM NaCl, 250 mM imidazole). The eluted proteins were further purified on protein concentrator (100 K MWCO) and diluted with storage buffer (150 mM NaCl, 20 mM HEPES, 0.1 mM EDTA, 1 mM DTT, 2% sucrose, and 20% glycerol). The concentration of purified protein was analyzed on SDS-PAGE gel.

**Preparation of target DNA and crRNA**. For single-particle tracking experiments, long DNA molecules (21 kb) were prepared by cutting λ-phage DNA (NEB, 48.5 kb in length) using restriction enzyme (*Eco*RI) and isolated by gel electrophoresis and extraction kit (utilized micron-sized bead) and purified using ethanol precipitation. Then, the purified DNA fragments of 21 kb were annealed with two biotinylated-oligos and one linker oligo (listed in Supplementary Table 1) for both ends modification and were ligated by T4 DNA ligase (NEB). Target site of AsCas12a RNP was located at a position that is 80% along the length of the DNA from the cosL side. The λ-phage DNA for drift rate measurement was annealed with biotinylated-oligos, which were complementary to cosL side of the λ-phage DNA (listed in Supplementary Table 1) and was ligated by T4 DNA ligase (NEB). For single-molecule FRET experiments, HPLC-purified short DNA strands were purchased from Integrated DNA Technologies (Coralville, IA) and some of them were labeled with Cy3 or Cy5 at the amine group of an internal amino modifier (dTC6). After then, DNA strands were annealed by slowly cooling down the mixture of the biotinylated strand and non-biotinylated strand in 1:2 molar ratio at 4 μM concentration in a buffer containing 10 mM Tris-HCl (pH 8.0) and 50 mM NaCl. For both single-molecule FRET and single-particle tracking experiments, HPLC-purified 42-nt crRNA strands were purchased from Integrated DNA

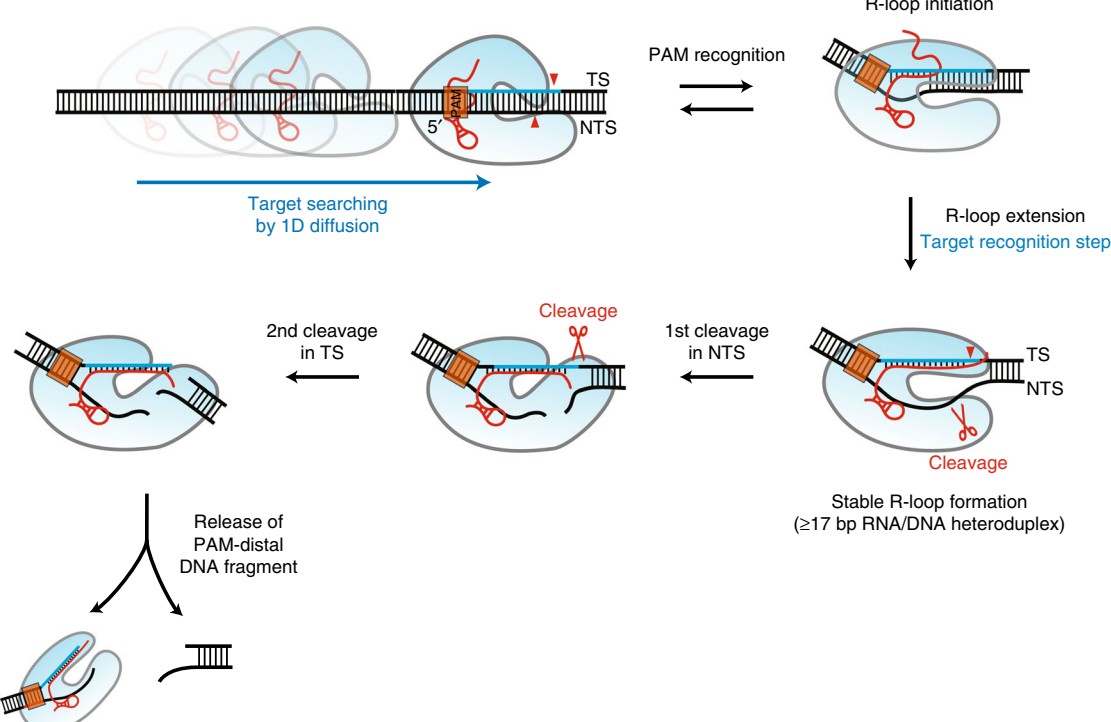

**Fig. 6** Proposed model for DNA target recognition and cleavage by AsCas12a RNP. An AsCas12a RNP first binds dsDNA and then diffuses along it (facilitated target searches including 1D diffusion). When an AsCas12a RNP encounters its target site, PAM-induced local unwinding by AsCas12a mediates DNA–RNA base pairing for checking complementarity (R-loop initiation). If the length of the R-loop is not sufficient (<17 bp), the AsCas12a RNP easily comes off the DNA (low stability under partial R-loop formation). Once a sufficiently long R-loop is formed (≥17 bp), the AsCas12a RNP strongly holds the DNA and is ready for DNA cleavage. After stable R-loop formation, AsCas12a sequentially cleaves the DNA, first the non-target strand and then the target strand, regardless of the presence of a PAM (PAM-independent sequential DNA cleavage). After successful DNA cleavage, AsCas12a quickly releases the PAM-distal DNA fragment but continues to hold the other fragment that contains both the PAM and protospacer

Technologies (USA) and some of them were labeled with Cy5 at the 3′ end of the strand.

**In vitro DNA cleavage assay by AsCas12a.** To prepare partially cognate DNA substrates, WT DNMT1 target sequences and partially cognate target sequences were cloned to pRG2 plasmid. DNA oligonucleotides for plasmid cloning used in this study are listed in Supplementary Table 2. Before in vitro DNA cleavage, target plasmids were linearized with restriction enzyme ScaI (New England Biolabs) then purified using a PCR cleanup kit (MGmed). An aliquot of 100 ng (4.4 nM) of linearized products was incubated with 100 ng (42.8 nM) purified AsCas12a protein and 35 ng (165.3 nM) DNMT1 crRNA (15 μl reaction in NEB buffer 3.1 at 37 °C for 30 min). After incubation, SDS contained loading dye was added and products were loaded in 1.5% agarose gel.

**Single-particle tracking experiments.** For the real-time tracking of AsCas12a–crRNA–Cy5 on the long DNA molecule, a custom flow chamber (2 mm channel width) was constructed with PEG-biotin-coated quartz surface[28]. Both side of 21 kb DNA were immobilized at PEG-biotin-coated quartz surface via biotin–streptavidin interaction by flowing the DNA at 50 μl/min of speed using syringe pump. Free 21 kb DNA molecules were stringently washed out by flowing T50 buffer (10 mM Tris-HCl, pH 8.0, 50 mM NaCl). Aliquots of 500 nM of AsCas12a and 1 μM of crRNA-Cy5 were preassembled for 5 min at room temperature (RT) in reaction buffer K100M10 (10 mM Tris-HCl, pH 8.0, 100 mM KCl, 10 mM MgCl$_2$, 1 mM DTT) just before the experiments. Final 1 nM of AsCas12a (with 2 nM of crRNA-Cy5) was introduced to the 21 kb DNA molecule in the chamber using syringe pump with reaction buffer K50M5 (10 mM Tris-HCl, pH 8.0, 50 mM KCl, 5 mM MgCl$_2$, 1 mM DTT). The oxygen scavenging system (0.8% (w/v) D-glucose (Sigma), 165 U/ml glucose oxidase (Sigma), 2170 U/ml catalase (Sigma), 3 mM Trolox) was used to minimize photobleaching and to suppress photoblinking of Cy5 fluorophore and Sytox orange (10 nM; Thermo Fisher Scientific). To observe Cy5-labeled AsCas12a–crRNA and Sytox orange-labeled dsDNA simultaneously, both 532 nm laser (Coherent, 100 mW) and 637 nm laser were exerted to the sample. Fluorescence signals were imaged in prism-type total internal reflection fluorescence microscopy (Olympus IX73, water type ×60 objective NA Aj1.2, Olympus)[51]. Emission signals collected by objective lens were

separated by dichroic mirror (FF640-FDi01, Semrock) and filtered (532 notch filter for Cy3, 642 notch filter for Cy5, chroma) and finally imaged at EMCCD camera (iXon DU-897D, Andor). Images were recorded using home-made software at 17–500 ms time resolution depending on the experiments. Almost all experiments were recorded in full frame (512 × 512 pixel), but 256 × 256 pixel images were recorded by 2 × 2 binning in case of requiring faster sampling rate (17 ms) for measuring diffusion coefficient and drift rate. Before imaging, temperature of flow chamber and buffer containing protein were set to ~37 °C by heating tube holder, prism, stage, and objective.

**Single-molecule FRET experiments.** To prevent the non-specific adsorption of proteins to the surface, cleaned quartz microscope slide and coverslip were coated with polyethylene glycol and biotinylated polyethylene glycol (Laysan Bio) in a 40:1 ratio[52]. A sample chamber was constructed between a quartz microscope slide and a coverslip using double-sided adhesive tape. DNA duplexes were immobilized on the PEG-coated surface via a streptavidin–biotin interaction. Single-molecule fluorescence images were taken in a home-built prism-type total internal-reflection microscope with 50 ms or 0.5 s time resolution. All measurements were performed at 37 °C with the following buffer: 10 mM Tris-HCl (pH 8.0), 100 mM NaCl, 10 mM MgCl$_2$, 1 mM DTT and the oxygen scavenging system (0.4% (w/v) glucose (Sigma), 1% (v/v) Trolox (Sigma), 1 mg/ml glucose oxidase (Sigma), 0.04 mg/ml catalase (Roche)) to reduce photobleaching of fluorophores[53], except the experiments for Figs. 4 and 5 (reaction buffer of K100M10 was used for AsCas12a and NEB buffer 3 for SpCas9). For kinetic analysis of initial binding of AsCas12a RNPs to immobilized target DNAs, new buffer containing 20 nM AsCas12a RNPs (unless stated otherwise) was infused in real time into the detection chamber by using a syringe pump (Fusion 200; Chemyx) while single-molecule images were being taken. Cy3 was excited by a green laser (532-nm, Sapphire; Coherent) for two-color FRET, and the alternative excitation of Cy3 and Cy5 was performed by a green and a red laser (637-nm, Obis; Coherent) using a mechanical shutter (LS3; Uniblitz). Direct excitation of Cy5 by a red laser confirmed the existence of Cy5-labeled cleaved DNA fragment or crRNA in FRET measurements. Fluorescence signals from Cy3 and Cy5 were collected using a water immersion objective lens (UPlanSApo ×60; Olympus), filtered through a 540-nm long-pass filter (LP03-532RU-25; Semrock) and a 633-nm notch filter (NF03-633E-25; Semrock) to reject scattered excitation laser lines, separated with a dichroic mirror (635dcxr; Chroma)

and imaged onto an electron-multiplying charged-coupled device camera (Ixon Ultra DU897U; Andor). For stable long-time FRET measurement in some experiments with 0.5-s time resolution, home-built autofocusing system was employed[54].

**Analysis of single-particle tracking data**. To analyze the diffusion behavior of AsCas12a–crRNA–Cy5, fluorescence image was analyzed using FIJI software[55]. To localize the position of AsCas12a–crRNA–Cy5, the intensity profile of AsCas12a RNP was fitted with 2D Gaussian profile using DiaTrack 3.05[56]. Tracking data were analyzed using Matlab 2016b (Mathworks) and OriginPro 8 (OriginLab) program. To calculate diffusion coefficient ($D_{coeff}$) of AsCas12a–crRNA–Cy5, the slope of mean-square displacement (MSD) versus time was fitted with the equation $MSD(t) = 2Dt$, where $t$ is the time interval. The MSD of each AsCas12a molecule was calculated by:

$$MSD(t) = \frac{1}{N}\sum_{i=1}^{N}(x_i(t) - x_i(0))^2 \qquad (1)$$

where $t = n\Delta T$ ($n = 1,2,3,\ldots$, $\Delta T$:Sampling rate). To calculate the drift rate, the step (displacement per frame) distributions of total trajectory of AsCas12a RNPs were fitted with Gaussian distribution resulting drift rate (nm/s) by multiplying sampling rate. We also calculated drift rates from 2step, 3step distribution and resulted final drift rate by averaging these.

**Analysis of single-molecule FRET data**. Time traces of Cy3 and Cy5 signals from a single molecule were extracted from a recorded movie using IDL software (ITT Visual Information Solutions) and analyzed using custom software written in MATLAB (MathWorks). To calculate the FRET efficiency, which is defined as the ratio of acceptor intensity to the sum of donor and acceptor intensities, data correction was performed for background noise and bleed-through of the donor signal to the acceptor channel. After data correction, we selected real molecule traces showing anti-correlations between the donor and acceptor or single-step photobleaching. To obtain the kinetic times from the time traces, FRET states for individual reaction steps were determined by the standard threshold method using fluorescence intensity and FRET time traces. Dwell time histograms of each state were fitted by an exponential decay function to obtain the corresponding kinetic times (Supplementary Figure 4).

**Data availability**. The data that support the findings of this study are available from the corresponding authors on reasonable request.

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

## Acknowledgements

We thank Prof. J.-B. Lee (POSTECH) for the helpful advice and discussion. This work was supported by National Research Foundation of Korea (NRF) Grants (No. 2017R1A6A3A01007932 to Yo.J.; 2016R1A6A3A11930747 to Y.H.C.; 2017R1A3B1023418 and 2018R1A2B2008995 to C.J.; 2015R1C1A1A01055164 and 2016R1A5A1007318 to S.L.; 2017M3A9G8084539 and 2018M3A9H3022412 to S.B.), and by a grant from the KIST Institutional Program to C.J., and by a grant from the National R&D Program for Cancer Control of Ministry (1520090), the GIST Research Insitute to S.L., and by grants from Next Generation BioGreen 21 Program (PJ01319301), Korea Healthcare technology R&D Project (HI16C1012) to S.B.

## Author contributions

C.J., S.L., and S.B. conceived this project.; Yo.J., J.G., and G.L. performed and analyzed the single-molecule tracking and FRET experiments in Figs. 1, 4, and 5 to elucidate the mechanism for target searching under the direction of I.-S.K. and C.J.; Y.H.C. and Yu.J. performed and analyzed single-molecule FRET experiments in Figs. 2 and 3 to elucidate the mechanism for target cleavage under the direction of S.L.; J.Y., Y.K.J., and S.H.L. prepared Cas variants and performed gel assay in Fig. 4c under the direction of J.-S.K. and S.B.; Yo.J., Y.H.C, Yu.J., C.J., S.L., and S.B. interpreted the data and wrote the manuscript.

## Additional information

**Competing interests:** The authors declare no competing interests.

