## [Peer Review File · Nature Communications]

Reviewers' comments:

Reviewer #1 (Remarks to the Author):

Jeon et al. investigate in this study the target recognition and DNA cleavage by Cas12a (Cpf1) using single-molecule fluorescence methods. They present a very comprehensive and impressive study that provides important insight into the dynamics of Cas12a when binding targets and subjecting them to cleavage. I generally support acceptance of this manuscript. The authors should, however, address several points that are detailed below.

1) The authors conclude that the enzyme uses a hopping mechanism during its 1D search of the target site. I generally dislike the term "hopping", since it is not defined what it actually means. I acknowledge that also published literature uses this term in an ambiguous way. In my opinion the authors can only conclude from their salt dependencies of the diffusion coefficient that salt bridges (or something similar) are important for the enzyme-DNA contact. This suggests that part of the interface is not too tight. Any interpretation beyond is quite speculative. Also the measurements in the presence of flow rule out larger displacements of the enzyme from the DNA during a hop. Otherwise it would be washed away.

The interpretation of the drift rate on the flow velocity is very misleading and should be adapted. The observed linear dependence is always expected from the Stokes-Einstein relation (connecting D to a friction coefficient) and is not dependent on the mechanism.

2) In figures 2e and 3 the authors provide histograms of the FRET efficiency without making clear whether these histogram were calculated only for the depicted time trajectory, from a collection of time trajectories or from mean FRET values calculated for a larger number of events. If the authors present these data only for a single particular time trace they should provide in addition histograms of mean FRET values for a larger number of events and always indicate from how many events the histograms were calculated

3) When determining cleavage from the FRET experiments the authors should explain how they can rule out or correct for photo bleaching that occurs at exponentially distributed times and will always affect the measured data. It maybe good to include additional data in which the photobleaching rates were characterized.

4) The authors should indicate in Figure 1E the total number of counts that entered the histogram and specify how the very large errors were obtained.

5) According to the Methods the authors do not correct their FRET values for quantum efficiency and detector sensitivity, such that they should label their graphs with "Apparent FRET efficiency" throughout.

6) The statement "Cas12a merely requires the PAM during DNA unwinding for R-loop initiation" is in my opinion misleading, since a permissive PAM is stringently required.

7) Line 232: 'Cyanine' should read 'Cyan'

Reviewer #2 (Remarks to the Author):

Jeon et al. use single-particle tracking and single-molecule FRET (smFRET) to investigate DNA-binding and cleavage by AsCas12a. Interestingly, they observe 1D diffusion of AsCas12a by single-particle tracking, in contrast to previous studies of SpCas9 (Sternberg et al. Nature, 2014). In smFRET experiments, the authors elegantly demonstrate that cleavage occurs in a sequential manner, and that AsCas12a may adopt different conformations following each catalytic step. They also provide evidence for differences in the PAM and target requirements between AsCas12a and

SpCas9, which could help to explain why the two proteins have different off-target activities in gene editing studies.

Overall, this is a well-performed and timely study that will be a great interest to a broad audience of readers interested in CRISPR biology, genome editing technology and nucleic acid-protein interactions. There are a few concerns that should be addressed prior to publication of this manuscript.

Major concerns:

1. The lack of observable "non-specific" binding in the smFRET experiment is a bit puzzling, in light of the apparent 1D diffusion shown in single-particle tracking experiments. Figure 1C appears to show several diffusion events at regions distal to the target, suggesting that AsCas12a can bind to non-specific regions of the DNA. If AsCas12a does indeed undergo 1D diffusion, wouldn't it be expected that short-range diffusion or hopping would be observable in the smFRET experiment as well? Do the authors have an explanation for why 1D diffusion was not detected in their smFRET experiments?

2. It would be helpful if more info was given on the number of traces analyzed the smFRET experiments (e.g. include number of traces in figure legends). In particular, how many traces were analyzed for Supplementary Figure 5? Were short (single frame) FRET events like the one at ~110 s observed in other traces? Is it possible that this represents very short-lived DNA sampling? Similar single-frame DNA sampling has been observed before for both SpCas9 (Singh et al. Nat Comm, 2016) and E. coli Cascade (Xue et al. Cell Rep, 2017) with "non-specific" constructs (i.e. lacking PAMs or targets).

Minor concerns

1. The peak height for the S4 events in Fig 2E is much lower than for S1-S3. Were S4 events rarer than the other types of events, and if so, is there an explanation for this observation?

2. The authors refer to the cleavage assay shown in Figure S4 as a gel shift assay at multiple points in the text and figure legend. This terminology is confusing as "gel shift" is normally used to refer to changes in mobility due to binding between a protein and nucleic acid, rather than cleavage of the nucleic acid. It would be more appropriate to refer to this as a cleavage assay analyzed by agarose gel electrophoresis. Also, the text says the cleavage assays were done on the same constructs as the smFRET experiments (p. 10, line 21), but the Online Methods says the cleavage assays were performed using linear plasmids. Should this say that the same target sequences were used?

3. The authors state (pp. 12-13, lines 266-268): "Nevertheless, to the best of our knowledge, no proof of a 1D diffusion mechanism during target searching has been reported for the CRISPR-Cas effectors (Cascade, Cas9, Cas12a) and this study may be the first direct evidence for that." There are two preprints on bioRxiv demonstrating 1D diffusion for Cascade (<https://doi.org/10.1101/208058>) and Cas9 (<https://doi.org/10.1101/264879>). It would be appropriate to cite these studies and change the text accordingly.

4. The authors state (p. 14, lines 304-305): "In the CRISPR-Cas system, the PAM sequence in the target DNA is critical for distinguishing self from non-self DNA [43]." Reference 43 is a study of the type III system, which does not utilize a PAM for self versus non-self discrimination. Consider citing PMID: 19246744, 18065545 and/or 21646539 instead.

Reviewer #3 (Remarks to the Author):

The CRISPR RNA-guided DNA endonucleases Cas9 and Cas12a (also known as Cpf1) are widely used in genome editing. Previous studies revealed that Cas12a has distinct features from those of Cas9, but the detailed mechanisms of target recognition and cleavage by Cas12a remained unclear. In this study, Jeon et al. used single-molecule fluorescence assays to directly monitor these processes of the Cas12a from *Acidaminococcus* sp. (AsCas12a). They determined that AsCas12a searches for the target site via 1D diffusion along the DNA molecule, and then induces the cleavage of the two DNA strands in a defined order. In addition, they found that the protospacer adjacent motif (PAM) makes only a limited contribution to DNA unwinding by AsCas12a, whereas the PAM is critical for the DNA unwinding by Cas9. Overall, this study has the potential to improve our mechanistic understanding of the CRISPR-Cas12a enzyme. I think that this manuscript will be suitable for publication in *Nature Communications*, if the following points are adequately addressed.

Points:

(1) P3, L55: The previous structures of Cas12a and a structural comparison with Cas12b suggested that both the TS and NTS are cleaved by the single active-site in the RuvC domain, with the Nuc domain guiding the TS to the RuvC active site. The information about the Cas12a-mediated cleavage mechanism should be provided in the Introduction.

(2) P9, L184: The authors should provide an explanation of why S4 was observed for "NTS-nick" (Fig. 3C), but not "TS-nick" (Fig. 3B). I think that this would reflect a conformational difference in the DNA prior to the release; i.e., the DNA folds back and interacts with the RuvC active site for "NTS-nick", as observed in the Cas12b structure, whereas it does not for "TS-nick" since the TS cleavage does not occur. Related to this, it would be better to modify the schematics in Fig. 3, so that the DNA substrate, rather than the RuvC domain, undergoes a conformational change for TS cleavage.

(3) P9, L189: The NTS cleavage by the single active-site in the RuvC domain was also proposed for FnCas12a (Swarts et al. *Mol. Cell* 2017). This should be cited.

(4) P10, L200: The R-loop formation and cleavage should be investigated for 14- and 16-nt matched substrates, to conclude that the 17-bp heteroduplex is required for the stable binding of AsCas12a.

(5) P13, L268: The sentences "A recent crystal structure¹³ revealed ..., including PAM recognition¹⁴ (Supplementary Figure 6B)." and Fig. S6A should be removed, since the hypothesis seems less likely. A structural comparison between LbCas12a-crRNA (Dong et al. *Nature* 2016) and LbCas12a-crRNA-DNA (Yamano et al. *Mol. Cell* 2017) showed that, unlike Cas9, which uses a groove for PAM recognition, Cas12a has a PAM-interacting channel, which is flexible and can undergo a closed-to-open conformational change upon DNA binding. Thus, it seems plausible that the DNA is accommodated within the PAM-interacting channel, rather than the central channel, for the 1D diffusion motion. This possibility should be discussed.

(6) P14, L298: Related to comment 2, it seems more likely that the DNA, rather than the RuvC domain, undergoes a structural change for TS cleavage, based on the previous studies of Cas12a and Cas12b. It would be informative if the authors could examine the conformational changes in the RuvC domain during the TS cleavage, using smFRET assays, where the Cas12a protein and the DNA are labeled, if technically possible.

(7) P15, L317: The sentence "The weaker AsCas12a RNP binding affinity for the PAM may provide insight into its higher target specificity" does not make sense. Instead, I think that the higher specificity of Cas12a, as compared to that of Cas9, is partly related to the authors' finding that,

unlike Cas9, Cas12a requires a longer RNA-DNA heteroduplex for binding and cleavage. This should be discussed.

(8) P15, L319: A previous study demonstrated that the specificity of Cas9 is improved by the enhanced proofreading (i.e., fewer mismatches are tolerated for the activation of the HNH domain), rather than simply the reduced affinity to the DNA substrate (Chen et al. Nature 2017). Thus, the sentence should be corrected.

Minor points:

(1) P7, L152: The reference should be cited.

(2) P3, L55 etc.: NUC domain -> Nuc domain ("NUC" lobe).

(3) Methods: There are some typos, and they should be fixed; e.g., P25, L545 etc.: 37 °C -> 37°C / P25, L548 etc.: 50mM -> 50 mM (put a space) / P25, L551: 4°C 13000 rpm -> 4°C at 13000 rpm / P26, L577: Assay -> assay / P28, L618: MgCl₂,1mM -> MgCl₂, 1 mM.

(4) P26, L580: The concentrations of Cas12a, RNA and DNA should also be provided as "nM" in addition to "ng".

(5) Supplementary Figure 4: It would be better to test the cleavage of the 14- and 16-nt matched targets as well. The term "gel shift assay" usually means the electrophoretic mobility shift assay (EMSA) to detect protein-DNA interactions, so this is confusing and should be corrected. The sizes of the molecular markers should be indicated on the left of the gel. The slight displacement of the red arrow (indicating the product) should be fixed.

(6) Supplementary Figure 5: 10nM -> 10 nM / 60 ms.. -> 60 ms.

Reviewers' comments:

Reviewer #1 (Remarks to the Author):

Jeon et al. investigate in this study the target recognition and DNA cleavage by Cas12a (Cpf1) using single-molecule fluorescence methods. They present a very comprehensive and impressive study that provides important insight into the dynamics of Cas12a when binding targets and subjecting them to cleavage. I generally support acceptance of this manuscript. The authors should, however, address several points that are detailed below.

Response: Thank you very much for the positive comments and the encouragement.

1) The authors conclude that the enzyme uses a hopping mechanism during its 1D search of the target site. I generally dislike the term "hopping", since it is not defined what it actually means. I acknowledge that also published literature uses this term in an ambiguous way. In my opinion the authors can only conclude from their salt dependencies of the diffusion coefficient that salt bridges (or something similar) are important for the enzyme-DNA contact. This suggests that part of the interface is not too tight. Any interpretation beyond is quite speculative. Also the measurements in the presence of flow rule out larger displacements of the enzyme from the DNA during a hop. Otherwise it would be washed away. The interpretation of the drift rate on the flow velocity is very misleading and should be adapted. The observed linear dependence is always expected from the Stokes-Einstein relation (connecting D to a friction coefficient) and is not dependent on the mechanism.

Response: Thank you for your valuable comment. We totally agree with the reviewer's comment that the term "hopping" is unclear and is likely to cause confusion. Therefore, we replaced the term with "the intermittent contact", instead. Next, we also agree with the reviewer's comment about the interpretation of the drift rate on the flow velocity. To avoid confusion and make the interpretation clearer, we rewrote the related sentences according to the reviewer's suggestion. Please refer the revised sentences (Page 6, Line 114).

2) In figures 2e and 3 the authors provide histograms of the FRET efficiency without making clear whether these histogram were calculated only for the depicted time trajectory, from a collection of time trajectories or from mean FRET values calculated for a larger number of events. If the authors present these data only for a single particular time trace they should provide in addition histograms of mean FRET values for a larger number of events and always indicate from how many events the histograms were calculated.

Response: Thank you for the comment. In figure 2E and 3, we collected all data points for each state from at least 45 time trajectories to obtain these FRET histograms. Following the reviewer's suggestion, we described data analysis method in detail, and provided the total number of time trajectories used in the analysis to the captions of Figure 2 and 3.

3) When determining cleavage from the FRET experiments the authors should explain how they can rule out or correct for photo bleaching that occurs at exponentially distributed times and will always affect the measured data. It maybe good to include additional data in which the photobleaching rates were characterized.

Response: Thank you for your valuable comment. We agree with the reviewer's concern regarding the effect of dye photobleaching during data analysis. First of all, we would like to point out that, in the original submission, we already confirmed that the disappearance of Cy5 signal occurring during the final intermediate (S4) in Figure 2D represents the release of cleavage product, not the dye photobleaching, by performing the experiment with doubly labelled target DNA (now revised to Supplementary Figure 3). In this experiment, we found that both signals from Cy5 fluorophores (labeled with crRNA and cleaved DNA fragment, respectively) and PIFE (protein induced fluorescence enhancement) effect on Cy3 disappeared at the same time. These findings strongly supports that the disappearance of Cy5 signal represents the release of crRNA, cleaved DNA fragment, and Cas12a protein as depicted in Figure 2B. In addition, to address the reviewer's suggestion, we measured the dye photobleaching time under the same condition as shown below and added the results in the revised manuscript (Supplementary figure2). In this measurement, we found that the dye photobleaching time (~1500-s) was about 15 times longer than the release time (~100-s). Thus, with all due respect, we believe that possible interference from dye photobleaching is not serious enough to reconsider data analysis.

Supplementary Figure 2. Characterization of dye photobleaching time in the experiment with 0.5-s time resolution

(A) Representative fluorescence intensity time trace showing a dye photobleaching event. In this measurement, the experimental condition including the excitation laser and the time resolution (0.5-s) was same as in Figure 2D. (B) Histogram of dye photobleaching time. The histogram was made from more than 500 bleaching events observed in three independent experiments with 0.5-s time resolution. The data was fit to single-exponential decay function to obtain the average dye photobleaching time (1558-s) under the experimental condition.

4) The authors should indicate in Figure 1E the total number of counts that entered the histogram and specify how the very large errors were obtained.

Response: Thank you for the comment. The total number of counts in Figure 1E was large enough ($N = 1054$) and the error bar was calculated by the standard error of the mean (s.e.m.) from 4 experiment sets. We added this additional information into the figure legend in the revised manuscript. Furthermore, to address the reviewer's suggestion, we changed the ordinate as the unit of "total number of counts", not "normalized count" to clearly specify the errors. With all due respect, we believe that the deviation is not serious enough to reconsider our interpretation about the PAM sequence-preferential binding of AsCas12a RNPs.

5) According to the Methods the authors do not correct their FRET values for quantum efficiency and detector sensitivity, such that they should label their graphs with "Apparent FRET efficiency" throughout.

Response: To address the reviewer's suggestion, we replaced the term of 'FRET efficiency' with 'Apparent FRET efficiency' in the revised manuscript.

6) The statement "Cas12a merely requires the PAM during DNA unwinding for R-loop initiation" is in my opinion misleading, since a permissive PAM is stringently required.

Response: Thank you for the comment. To address the reviewer's concern and clarify the meaning, we revised the sentence that Cas12a plays a negligible role after DNA unwinding for R-loop initiation. Please refer the revised sentences (Page 15, Line 315).

7) Line 232: 'Cyanine' should read 'Cyan'

Response: We have corrected it in the revised manuscript.

Reviewer #2 (Remarks to the Author):

Jeon et al. use single-particle tracking and single-molecule FRET (smFRET) to investigate DNA-binding and cleavage by AsCas12a. Interestingly, they observe 1D diffusion of AsCas12a by single-particle tracking, in contrast to previous studies of SpCas9 (Sternberg et al. Nature, 2014). In smFRET experiments, the authors elegantly demonstrate that cleavage occurs in a sequential manner, and that AsCas12a may adopt different conformations following each catalytic step. They also provide evidence for differences in the PAM and target requirements between AsCas12a and SpCas9, which could help to explain why the two proteins have different off-target activities in gene editing studies.

Overall, this is a well-performed and timely study that will be a great interest to a broad audience of readers interested in CRISPR biology, genome editing technology and nucleic acid-protein interactions. There are a few concerns that should be addressed prior to publication of this manuscript.

Response: Thank you very much for the positive comments and the encouragement.

Major concerns:

1. The lack of observable "non-specific" binding in the smFRET experiment is a bit puzzling, in light of the apparent 1D diffusion shown in single-particle tracking experiments. Figure 1C appears to show several diffusion events at regions distal to the target, suggesting that AsCas12a can bind to non-specific regions of the DNA. If AsCas12a does indeed undergo 1D diffusion, wouldn't it be expected that short-range diffusion or hopping would be observable in the smFRET experiment as well? Do the authors have an explanation for why 1D diffusion was not detected in their smFRET experiments?

Response: Thank you for your valuable comment. We totally agree with the reviewer's concern that AsCas12a can bind to non-specific regions of the DNA in light of the apparent

1D diffusion shown in our single-particle tracking experiments. However, we would point out that these non-specific binding to short DNA (smFRET) is expected to be a highly transient event in the several hundreds of micro-seconds range (~100 μ s) by considering our estimation using the diffusion coefficient, D_{coeff} (1.73 μ m²/s), in our 1D diffusion data. The diffusing time of Cas12a on short DNA substrate can be approximately calculated as follows. The 1D diffusion of protein on DNA is described by its mean square displacement, $\langle x^2 \rangle = 2Dt$. Where $\langle x^2 \rangle$ is an average (squared) distance travelled by the protein on DNA, D is a diffusion coefficient, and t is a duration of the random walk. The DNA length used in our experiments is 17.34 nm (51bp), and the long travel length (not average) is approximately set as 34.68 nm, because one end of DNA is blocked by the surface via biotin and avidin interaction. As a result, the diffusion time of Cas12a on the short DNA is about 350 μ s. Therefore, we think that it was very hard to detect such highly transient short diffusion events through our single molecule FRET measurements with a 50-ms time resolution (which is the time limit of our instrument).

2. It would be helpful if more info was given on the number of traces analyzed the smFRET experiments (e.g. include number of traces in figure legends). In particular, how many traces were analyzed for Supplementary Figure 5? Were short (single frame) FRET events like the one at ~110 s observed in other traces? Is it possible that this represents very short-lived DNA sampling? Similar single-frame DNA sampling has been observed before for both SpCas9 (Singh et al. Nat Comm, 2016) and E. coli Cascade (Xue et al. Cell Rep, 2017) with “non-specific” constructs (i.e. lacking PAMs or targets).

Response: Thank you for the comment. As the reviewer suggested, we provided the number of traces used in the analysis to the caption of figures if necessary, including Supplementary Figure 5 (now revised to Supplementary Figure 6). Regarding the reviewer’s comment about short FRET events shown in the trace in Supplementary Figure 6, it may represent a very short-lived binding event to non-specific regions of the DNA. In addition, to address the reviewer’s comment, we counted the short event by eyes and found 167 events for 200 traces. We would like to emphasize that, as we mentioned above (the response to comment #1), the non-specific binding events are expected to have very short binding lifetime; thus the short FRET events for non-specific constructs were rarely detected in our measurements.

Minor concerns

1. The peak height for the S4 events in Fig 2E is much lower than for S1-S3. Were S4 events rarer than the other types of events, and if so, is there an explanation for this

observation?

Response: Thank you for the comment. As the reviewer pointed out, our data in Figure 2E is likely to cause confusion. In our measurements, each state, except for S1, occurs only once during a single catalytic cycle. As described in the above response (for comment #2 by the reviewer 1), we collected all data points for each state from a number of time trajectories to obtain these FRET histograms. Thus, the lower peak height for the S4 events in Figure 2E reflects that S4 events have lower dwell time than the other types of events, rather than less frequent. To avoid confusion, we added more explanation for data analysis to the caption of Figure 2 in the revised manuscript.

2. The authors refer to the cleavage assay shown in Figure S4 as a gel shift assay at multiple points in the text and figure legend. This terminology is confusing as “gel shift” is normally used to refer to changes in mobility due to binding between a protein and nucleic acid, rather than cleavage of the nucleic acid. It would be more appropriate to refer to this as a cleavage assay analyzed by agarose gel electrophoresis. Also, the text says the cleavage assays were done on the same constructs as the smFRET experiments (p. 10, line 21), but the Online Methods says the cleavage assays were performed using linear plasmids. Should this say that the same target sequences were used?

Response: Thank you for the comment. As the reviewer suggested, we changed the term of “gel shift” to “*in vitro* DNA cleavage assay”. Furthermore, we apologize for the unclear expression of DNA constructs. For the *in vitro* DNA cleavage assay, we used DNA plasmids containing the same target sequences to the smFRET experiments. We corrected the sentence in the revised manuscript (Page 10, Line 213).

3. The authors state (pp. 12-13, lines 266-268): “Nevertheless, to the best of our knowledge, no proof of a 1D diffusion mechanism during target searching has been reported for the CRISPR-Cas effectors (Cascade, Cas9, Cas12a) and this study may be the first direct evidence for that.” There are two preprints on bioRxiv demonstrating 1D diffusion for Cascade (<https://doi.org/10.1101/208058>) and Cas9 (<https://doi.org/10.1101/264879>). It would be appropriate to cite these studies and change the text accordingly.

Response: Thank you for the comment. This genome editing field moves so fast that we missed those studies. We corrected the sentence that we first suggest 1D diffusion only for CRISPR-Cas12a nucleases. (Page 13, Line 269)

4. The authors state (p. 14, lines 304-305): “In the CRISPR-Cas system, the PAM sequence in the target DNA is critical for distinguishing self from non-self DNA [43].” Reference 43 is a study of the type III system, which does not utilize a PAM for self versus non-self discrimination. Consider citing PMID: 19246744, 18065545 and/or 21646539 instead.

Response: Thank you for the comment. Following the reviewer’s suggestion, we corrected the references in the revised manuscript.

Reviewer #3 (Remarks to the Author):

The CRISPR RNA-guided DNA endonucleases Cas9 and Cas12a (also known as Cpf1) are widely used in genome editing. Previous studies revealed that Cas12a has distinct features from those of Cas9, but the detailed mechanisms of target recognition and cleavage by Cas12a remained unclear. In this study, Jeon et al. used single-molecule fluorescence assays to directly monitor these processes of the Cas12a from *Acidaminococcus* sp. (*AsCas12a*). They determined that *AsCas12a* searches for the target site via 1D diffusion along the DNA molecule, and then induces the cleavage of the two DNA strands in a defined order. In addition, they found that the protospacer adjacent motif (PAM) makes only a limited contribution to DNA unwinding by *AsCas12a*, whereas the PAM is critical for the DNA unwinding by Cas9. Overall, this study has the potential to improve our mechanistic understanding of the CRISPR-Cas12a enzyme. I think that this manuscript will be suitable for publication in *Nature Communications*, if the following points are adequately addressed.

Response: Thank you very much for the positive comments and the encouragement.

Points:

(1) P3, L55: The previous structures of Cas12a and a structural comparison with Cas12b suggested that both the TS and NTS are cleaved by the single active-site in the RuvC domain, with the Nuc domain guiding the TS to the RuvC active site. The information about the Cas12a-mediated cleavage mechanism should be provided in the Introduction.

Response: Thank you for the comment. To address the reviewer’s suggestion, we added the recent structural results briefly in the INTRODUCTION part. We would point out that the more detailed descriptions are denoted in the DISCUSSION part. (Page 3, Line 50)

(2) P9, L184: The authors should provide an explanation of why S4 was observed for “NTS-nick” (Fig. 3C), but not “TS-nick” (Fig. 3B). I think that this would reflect a conformational difference in the DNA prior to the release; i.e., the DNA folds back and interacts with the RuvC active site for “NTS-nick”, as observed in the Cas12b structure, whereas it does not for “TS-nick” since the TS cleavage does not occur. Related to this, it would be better to modify the schematics in Fig. 3, so that the DNA substrate, rather than the RuvC domain, undergoes a conformational change for TS cleavage.

Response: Thank you for your valuable comment. As the reviewer suggested, this difference shown in these two constructs, ‘NTS-nick’ and ‘TS-nick’, appears to reflect that a large conformational change in the DNA is required for target strand cleavage. Following the reviewer’s suggestion, we discussed this point in the revised manuscript (Page 9, Line 184). However, it has not yet been clarified whether the conformational change is undergone in DNA substrate or the RuvC domain of the protein or both of them. To address this issue, smFRET experiments using fluorescently labeled protein should be performed. However, this experiment requires a long experimental time due to the difficulty in preparation of fluorescently labeled protein. Therefore, we would like to avoid the mistake of making a hasty generalization at the present time; thus, we would prefer not to modify the schematics in Fig. 3 and to address this issue in the future with a more comprehensive study. And we believe that this issue would be one of major contents in another project.

(3) P9, L189: The NTS cleavage by the single active-site in the RuvC domain was also proposed for FnCas12a (Swarts et al. Mol. Cell 2017). This should be cited.

Response: We have added the reference in the revised manuscript.

(4) P10, L200: The R-loop formation and cleavage should be investigated for 14- and 16-nt matched substrates, to conclude that the 17-bp heteroduplex is required for the stable binding of AsCas12a.

Response: Thank you for the comment. To address the reviewer’s suggestion, we measured the binding and cleavage activities of AsCas12a on 14- and 16-bp matched DNA templates additionally. Consequently, we measured that the 17-bp heteroduplex is still required for the complete stable binding as shown below and this observation was in line with the conclusion in the initial manuscript. We revised the Figure 4C and Supplementary Figure 4 (now revised to Supplementary Figure 5).

Revised Figure 4C.

(5) P13, L268: The sentences “A recent crystal structure¹³ revealed ..., including PAM recognition¹⁴ (Supplementary Figure 6B).” and Fig. S6A should be removed, since the hypothesis seems less likely. A structural comparison between LbCas12a-crRNA (Dong et al. Nature 2016) and LbCas12a-crRNA-DNA (Yamano et al. Mol. Cell 2017) showed that, unlike Cas9, which uses a groove for PAM recognition, Cas12a has a PAM-interacting channel, which is flexible and can undergo a closed-to-open conformational change upon DNA binding. Thus, it seems plausible that the DNA is accommodated within the PAM-interacting channel, rather than the central channel, for the 1D diffusion motion. This possibility should be discussed.

Response: Thank you for the comment. We agree with the reviewer’s comment that our speculation was not convincing. To address the reviewer’s suggestion, we removed the related sentences and Supplementary Figure 6, and totally rewrote it based on the recent structural studies. (Page 13, Line 270). To elucidate this issue, more complicated experimental design, such as fluorescence dye labeling of Cas12a protein, will be needed, but, this issue requires long-term experiments due to the difficulty in preparation of fluorescently labeled proteins. Thus, with all due respect, we would like to address this issue in the future with a more comprehensive study. And we believe that this issue would be one of major contents in another project.

(6) P14, L298: Related to comment 2, it seems more likely that the DNA, rather than the

RuvC domain, undergoes a structural change for TS cleavage, based on the previous studies of Cas12a and Cas12b. It would be informative if the authors could examine the conformational changes in the RuvC domain during the TS cleavage, using smFRET assays, where the Cas12a protein and the DNA are labeled, if technically possible.

Response: Thank you for the comment. We totally agree with the reviewer's comment. For site-specific dye labeling, however, we should prepare a cysteine-less mutant of AsCas12a (WT enzyme has as many as 8 natural cysteines) and then introduce new cysteine residues, which will be labeled with maleimide dyes, into specific positions. Furthermore, following each mutational step, enzymes should be checked by biochemical assays to determine whether catalytic activity is conserved. Hence, this issue requires long-term experiments due to the difficulty in preparation of fluorescently labeled protein. Thus, with all due respect, we would like to address this issue in the future with a more comprehensive study. And we believe that this issue would be one of major contents in another project.

(7) P15, L317: The sentence "The weaker AsCas12a RNP binding affinity for the PAM may provide insight into its higher target specificity" does not make sense. Instead, I think that the higher specificity of Cas12a, as compared to that of Cas9, is partly related to the authors' finding that, unlike Cas9, Cas12a requires a longer RNA-DNA heteroduplex for binding and cleavage. This should be discussed.

Response: Thank you for the comment. We agree with the reviewer's comment and revised the sentence according to the reviewer's suggestion that the longer seed sequences for stable binding of AsCas12a RNP may provide insight into its higher target specificity. (Page 15, Line 318)

(8) P15, L319: A previous study demonstrated that the specificity of Cas9 is improved by the enhanced proofreading (i.e., fewer mismatches are tolerated for the activation of the HNH domain), rather than simply the reduced affinity to the DNA substrate (Chen et al. Nature 2017). Thus, the sentence should be corrected.

Response: Thank you for the comment. And we apologize for the inaccurate reference. We think that our findings can support the previous studies that improved the Cas9 specificity by reducing its binding affinity for target DNA. Thus, it would be better to remove the reference, Chen et al. Nature 2017.

Minor points:

(1) P7, L152: The reference should be cited.

Response: We have added the reference in the revised manuscript.

(2) P3, L55 etc.: NUC domain -> Nuc domain (“NUC” lobe).

Response: We have corrected the term in the revised manuscript.

(3) Methods: There are some typos, and they should be fixed; e.g., P25, L545 etc.: 37 °C -> 37°C / P25, L548 etc.: 50mM -> 50 mM (put a space) / P25, L551: 4°C 13000 rpm -> 4°C at 13000 rpm / P26, L577: Assay -> assay / P28, L618: MgCl₂,1mM -> MgCl₂, 1 mM.

Response: We have corrected all of them as possible in the revised manuscript.

(4) P26, L580: The concentrations of Cas12a, RNA and DNA should also be provided as “nM” in addition to “ng”.

Response: We have provided the concentrations as the unit of “nM” additionally in the revised manuscript.

(5) Supplementary Figure 4: It would be better to test the cleavage of the 14- and 16-nt matched targets as well. The term “gel shift assay” usually means the electrophoretic mobility shift assay (EMSA) to detect protein-DNA interactions, so this is confusing and should be corrected. The sizes of the molecular markers should be indicated on the left of the gel. The slight displacement of the red arrow (indicating the product) should be fixed.

Response: Thank you for the comment. To address the reviewer’s suggestion, we changed the term of “gel shift assay” to “*in vitro* DNA cleavage assay”. In addition, we measured the cleavage activities of AsCas12a on 14- and 16-bp matched DNA templates additionally and added the information such as the sizes of the marker as shown below.

Supplementary Figure 5. *In vitro* DNA cleavage assay for determining DNA cleavage efficiency of AsCas12a with partially cognate DNAs using agarose gel electrophoresis

DNA cleavage efficiencies of AsCas12a with partially cognate DNAs are assessed by *in vitro* cleavage assay using agarose gel electrophoresis (Online Methods). Number of matched nucleotides with crRNA are extending from the PAM-proximal side. Substrate and cleaved products are indicated with blue and red arrows, respectively. The experiment was repeated three times. Statistics are shown in Figure 4C.

(6) Supplementary Figure 5: 10nM -> 10 nM / 60 ms.. -> 60 ms.

Response: We have corrected them in the revised manuscript.

REVIEWERS' COMMENTS:

Reviewer #1 (Remarks to the Author):

The authors fully addressed my previously made remarks.

Reviewer #2 (Remarks to the Author):

The manuscript by Jeon et al. provides a thorough investigation of DNA target search, binding and cleavage by Cas12a. The authors' revisions have satisfied my minor concerns, and the manuscript is now suitable for publication. I would like to note that another single-molecule FRET study of Cas12a was published recently in PNAS (Singh et al. PNAS, 2018). However, I do not think that paper greatly affects the impact of this manuscript. Although the two papers overlap in some respects, Jeon et al. includes many valuable insights that are not present in the Singh et al. study. These include the observation of 1D diffusion by Cas12a, the order of product cleavage by Cas12a and the evidence for alternative conformations during target binding and cleavage. Therefore, I believe this manuscript is suitable for publication in Nature Communications, and will still be of great interest to a broad audience.

One minor suggestion:

P 15 line 321: the newly added text should likely read "the PAM plays a negligible role for Cas12a after DNA unwinding for R-loop initiation."

Reviewer #3 (Remarks to the Author):

I feel that the authors addressed the points raised in the previous round of review.

Point-by-point response to the reviewer #2:

Reviewer #2 (Remarks to the Author):

The manuscript by Jeon et al. provides a thorough investigation of DNA target search, binding and cleavage by Cas12a. The authors' revisions have satisfied my minor concerns, and the manuscript is now suitable for publication. I would like to note that another single-molecule FRET study of Cas12a was published recently in PNAS (Singh et al. PNAS, 2018). However, I do not think that paper greatly affects the impact of this manuscript. Although the two papers overlap in some respects, Jeon et al. includes many valuable insights that are not present in the Singh et al. study. These include the observation of 1D diffusion by Cas12a, the order of product cleavage by Cas12a and the evidence for alternative conformations during target binding and cleavage. Therefore, I believe this manuscript is suitable for publication in Nature Communications, and will still be of great interest to a broad audience.

Response: We have added the reference (Singh et al. PNAS, 2018) in the revised manuscript.

One minor suggestion:

P 15 line 321: the newly added text should likely read “the PAM plays a negligible role for Cas12a after DNA unwinding for R-loop initiation.”

Response: We have corrected the sentence as the reviewer suggested to below.

“unlike type I Cascade and type II Cas9, the PAM of Cas12a plays a negligible role after DNA unwinding for R-loop initiation.”